# Angiopoietin-2: A Therapeutic Target for Vascular Protection in Hutchinson–Gilford Progeria Syndrome

**DOI:** 10.3390/ijms252413537

**Published:** 2024-12-18

**Authors:** Sahar Vakili, Kan Cao

**Affiliations:** Department of Cell Biology and Molecular Genetics, University of Maryland, College Park, MD 20742, USA; saharv@umd.edu

**Keywords:** Angiopoietin-2, aging, therapeutic potential, CVD, HGPS

## Abstract

Hutchinson–Gilford progeria syndrome (HGPS) is a pediatric condition characterized by clinical features that resemble accelerated aging. The abnormal accumulation of a toxic form of the lamin A protein known as progerin disrupts cellular functions, leading to various complications, including growth retardation, loss of subcutaneous fat, abnormal skin, alopecia, osteoporosis, and progressive joint contractures. Death primarily occurs as the result of complications from progressive atherosclerosis, especially from cardiac disease, such as myocardial infarction or heart failure, or cerebrovascular disease like stroke. Despite the availability of lonafarnib, the only US Food and Drug Administration-approved treatment for HGPS, cardiovascular complications remain the leading cause of morbidity and mortality in affected patients. Defective angiogenesis—the process of forming new blood vessels from existing ones—plays a crucial role in the development of cardiovascular disease. A recent study suggests that Angiopoietin-2 (Ang2), a pro-angiogenic growth factor that regulates angiogenesis and vascular stability, may offer therapeutic potential for the treatment of HGPS. In this review, we describe the clinical features and key cellular processes impacted by progerin and discuss the therapeutic potential of Ang2 in addressing these challenges.

## 1. Introduction

Hutchinson–Gilford progeria syndrome (HGPS) is a rare, fatal genetic disorder caused by a de novo point mutation in exon 11 of the *LMNA* gene (c.1824C > T; p.G608G) [1]. This mutation produces a toxic variant of lamin A, called progerin, with a 50-amino-acid deletion near its C terminus [1,2]. As a result of this deletion, progerin cannot undergo normal lamin A processing and remains permanently farnesylated [3]. Farnesylated progerin accumulates in the nuclear lamina, leading to nuclear abnormalities, genomic instability, altered redox homeostasis, and disrupted gene expression [3,4]. In normal aging, progerin levels also increase due to sporadic use of the cryptic splice site in exon 11 [5,6]. Children with HGPS appear normal at birth; however, they begin to develop symptoms resembling accelerated physiological aging within the first year of life. This condition is characterized by clinical features such as alopecia, osteoporosis, abnormal skin, loss of subcutaneous fat, joint contractures, and progressive atherosclerosis [4,7]. Complications from severe atherosclerosis, such as myocardial infarction, heart failure, and cerebrovascular disease (stroke), are the main cause of death in these patients [7]. Although *lonafarnib*, a farnesyltransferase inhibitor, extends the lifespan and quality of life of HGPS patients, there is currently no cure or effective treatment to prevent the onset of aging or reverse the negative outcomes associated with this condition.

The healthy endothelium is a continuous simple monolayer that lines the inner surface of blood vessels and can respond to chemical and physical signals to maintain vascular hemostasis and vascular tone [8]. Various physical and biochemical insults can alter endothelial cell (EC) function, contributing to the development of atherosclerosis [9]. Endothelial dysfunction in HGPS can be a prominent contributor to the development of cardiovascular diseases (CVDs) [10]. EC dysfunction also occurs during the normal aging process and is considered an independent risk factor for the development of many vasculopathies [11]. In addition to atherosclerosis and coronary artery disease, EC dysfunction in older adults could lead to other age-associated diseases such as renal dysfunction, osteoporosis, retinopathy, Alzheimer’s disease, pulmonary hypertension, macular degeneration, and stroke [12,13,14,15,16]. Therefore, strategies to enhance EC health are essential for promoting overall well-being, especially in aging populations.

A recent study identified Angiopoietin-2 (Ang2) as a missing growth factor in the progeria vasculature and demonstrated that exogenous treatment with this factor reversed EC dysfunction in vitro [17]. The angiopoietins (Ang) are a family of four secreted growth factors, Ang1, Ang2, Ang3, and Ang4, that bind to the endothelial receptor tyrosine kinase Tie2 and regulate vascular development and function [18,19]. Of the four angiopoietins, *Ang1* and *Ang2* are the best characterized. Ang1 is a pro-angiogenic growth factor that activates Tie2, whereas Ang2 was initially identified as a Tie2 antagonist with disruptive activity toward blood vessels [19].

Emerging studies have demonstrated that Ang2 has context-dependent agonistic activity and can activate the Tie2 receptor and induce endothelial cell survival, depending on the cell type and context [20,21,22]. While different cell types express Ang1, Ang2 expression is limited to endothelial cells [19]. Ang2 was first identified by homology screening following the identification of Ang1, with 60 percent similarity [18]. It is a 496-amino-acid-long protein with an NH2-terminal coiled-coil domain, a secretion signal peptide, and a COOH-terminal fibrinogen-like domain (Figure 1). Ang2 is stored in Weibel–Palade bodies (WPBs); it acts in an autocrine manner, and its expression is highly regulated [19].

Ang2 has a multifaceted role in regulating vascular remodeling, which can result in either vessel formation or regression depending on how its expression interacts with other angiogenic signals [23]. In clinical practice, Ang2 is often studied in the context of diseases characterized by endothelial dysfunction, such as cardiovascular diseases, diabetes, and cancer.

The angiopoietin/Tie2/Akt signaling pathway plays a crucial role in maintaining EC function during angiogenesis. Under normal conditions, activation of Tie2 enhances vascular integrity by promoting EC survival, improving cellular junctions, and reducing the response to inflammation [24]. Higher concentrations of Ang2 (800 ng/mL) have been shown to induce Tie2 phosphorylation, which activates PI3K at the p85 subunit. This, in turn, phosphorylates Akt at Ser473, promoting EC proliferation and cell survival [21]. Additionally, Tie2 phosphorylation mediated by Ang2 also leads to chemotaxis and tube formation in murine brain capillary ECs. Chemotaxis improvement was associated with c-Fes activation through PI3K activation, while tube formation was linked to c-Fyn activation, independently of PI3K [25].

Although the Tie2/Akt signaling pathway has not been extensively studied in the context of vascular disease in progeria, emerging evidence suggests that Akt pathway dysregulation plays a significant role in the severe growth retardation observed in these patients. The mislocalization of progerin outside the nucleus interacts with insulin-like growth factor receptor 1 (IGFR-1), thereby impairing the IGF-1/AKT signaling pathway. While Akt deficiency in Zmpste24^−/−^ mice induces the aging phenotype, increasing Akt activity has been shown to significantly improve postnatal growth and lifespan in these animals [26]. In line with this, another study demonstrated that reducing the activity of isoprenylcysteine carboxyl methyltransferase (ICMT) activates the Akt signaling pathway and eliminates the disease phenotype in Zmpste24-deficient mice [27]. Furthermore, gene expression analysis comparing HGPS and normal controls identified differentially expressed genes that were significantly enriched in the PI3K-Akt signaling pathway [28]. All these studies provide strong evidence that targeting this pathway appears promising for the treatment of progeria. Given Ang2’s crucial role in regulating angiogenesis and maintaining a balance between vessel growth and stability through Akt activation, this review aims to discuss the potential therapeutic benefits of Ang2 treatment on clinical complications associated with HGPS.

## 2. Potential Therapeutic Roles of Angiopoietin-2 in HGPS

### 2.1. Atheroprotective Role of Ang2

Although most HGPS patients lack classical cardiovascular risk factors, they typically die from complications of atherosclerosis at an average age of 14.6 years [29]. This suggests that progerin expression leads to pathological alteration in the arterial wall. Supporting these data, autopsies of HGPS patients who died from heart failure or acute myocardial infarction revealed varying levels of generalized atherosclerosis in their major arteries [30]. Furthermore, another case review of 12 autopsies demonstrated that cardiovascular complications in HGPS patients were mainly due to coronary artery occlusion caused by atherosclerosis rather than cardiomyopathy [31].

Notably, all cases showed calcification of the aortic vessels, along with mitral valves, which contributed to the formation of atherosclerotic plaques in the aorta [31]. In addition, arterial stenosis serves as an early sign of atherosclerotic plaque formation in HGPS patients [32]. Common characteristics of atherosclerosis observed in both normal aging and in patients with HGPS include inflammation, loss of vascular smooth muscle, and the formation, erosion, and rupture of plaques [4,33]. Moreover, HGPS patients distinctly exhibit adventitial thickening and fibrosis [4].

Atherosclerosis is driven by a combination of hypercholesterolemia and vascular inflammation [34]. The endothelium plays a vital role in the prevention of atherosclerosis, and its dysfunction is crucial in the development of the disease [35]. In a study, Ahmed et al. explored the role of Ang2 in suppressing atherosclerosis in apoE^−/−^ mice [36]. They reported that overexpression of Ang2 decreased lesion size, reduced macrophage accumulation, and lowered oxidized LDL content in plaques of atherosclerosis-prone apoE^−/−^ mice [36]. This protective effect was lost upon endothelial cell nitric oxide synthase (eNOS) inhibition, suggesting that nitric oxide (NO) produced by Ang2 may contribute to atheroprotection [36]. This study suggested that Ang2 inhibits atherosclerotic lesion development partly by reducing LDL oxidation and macrophage accumulation through the activation of eNOS [36].

These findings were further confirmed in vitro, where Ang2 treatment significantly reduced LDL oxidation in human umbilical vein endothelial cells (HUVECs) [36]. This effect was partially abolished by NG-nitro-L-arginine methyl ester (L-NAME), an inhibitor of eNOS, indicating that LDL oxidation inhibition occurs through a NO-dependent mechanism [36]. The loss of eNOS activity is a well-established feature of HGPS endothelial cells, and NO derived from the endothelium appears to play a crucial role in preventing atherosclerosis [37,38]. Vakili et al. demonstrated that treating HGPS ECs with Ang2 led to increased eNOS activity and enhanced production and release of NO [17]. Although other growth factors, such as VEGF and Ang1, can also stimulate the release of NO, these effects are often accompanied by the recruitment of inflammatory cells [39,40,41].

Additionally, VEGF was found to increase plaque formation in double-deficient apoE/apoB100 mice, while Ang1 was ineffective in protecting against the development of arteriosclerosis in rat cardiac allografts [42,43]. Another study found that administering recombinant Ang2 significantly inhibited angiotensin II-induced aortic dilation in apoE-deficient mice [44]. This treatment not only protected against suprarenal aorta (SRA) rupture but also prevented the development of atherosclerosis in the aortic arch [44]. Mice treated with recombinant Ang2 exhibited significantly lower levels of MCP-1 and IL-6 in their plasma, indicating that Ang2 may protect against inflammation [44]. The protective effect of Ang2 administration in this study was demonstrated by its ability to induce Tie2 activation and decrease monocyte/macrophage infiltration within the aortic wall in the presence of angiotensin II [44].

Given that Ang2 treatment induces NO production by enhancing eNOS activity, it may mitigate endothelial dysfunction, a primary contributor to atherosclerosis, in the context of HGPS. Additionally, by lowering LDL oxidization, Ang2 may prevent the progression of atherosclerosis in these patients by reducing vascular inflammation and plaque formation. Furthermore, the ability of Ang2 to decrease macrophage infiltration in atherosclerotic lesions may also aid in plaque stabilization in HGPS. The anti-inflammatory effects of Ang2 may contribute to more stable vessels, thereby postponing the progression of atherosclerosis. Ultimately, Ang2 treatments enhance the EC function and health by activating the Tie receptor, which can reduce macrophage and monocyte infiltration, further contributing to overall atheroprotection.

### 2.2. Alleviating Endothelial Dysfunction

The presence of endothelial dysfunction in HGPS patients is associated with cardiovascular diseases, particularly atherosclerosis. One of the major roles of healthy endothelium is maintaining the balance between production and detoxification of the reactive oxygen species [45]. Experimental and clinical studies on atherosclerosis suggested that oxidative stress, through various mechanisms, can lead to endothelial dysfunction [46,47].

Exogenously expressed progerin in primary cultures of human coronary endothelial cells induces oxidative stress, accompanied by increased inflammation, with persistent DNA damage and elevated expression of cell cycle arrest proteins [10]. Several studies suggested that mechanical stress influences EC function and that the initiation of pro-inflammatory, pro-atherogenic, and pro-thrombotic responses increases the risk of endothelial dysfunction [48,49,50]. Progerin expression in ECs can also impair their mechanoresponse and contribute to cardiovascular pathology [37]. Changes in the mechanosensitive elements, such as F-actin/G-actin ratios on the nuclear envelope of the ECs induced by progerin, lead to this abnormal mechanosensing.

Impairment of angiogenic processes is associated with EC dysfunction, which compromises the integrity of the circulatory system [51]. It is known that EC dysfunction and impaired nitric oxide bioavailability play critical roles in angiogenic incompetence in patients with progeria [37]. Progerin impairs vasculogenesis by inhibiting the activity of matrix metalloproteinase MMP2 and MMP9 while increasing the expression of tissue inhibitors of metalloproteinase TIMP1 and TIMP2 [37]. A decreased wound healing rate in progerin-expressing ECs under shear stress is another indication of endothelial dysfunction [52].

Increased inflammatory signaling in ECs is also considered one of the initial steps in the development of atherosclerosis [53]. EC senescence is a hallmark of progeria syndrome, and patient-derived ECs exhibit telomere erosion, increased β-gal staining, DNA damage, and elevated expression of the well-known senescent markers p16 and p21 compared to normal ECs [54,55,56]. Reduced expression of eNOS in response to shear stress is another mechanism involved in EC dysfunction associated with HGPs [37,38].

Activation of the Tie2 receptor is crucial for endothelial cell survival, vascular maturation, and integrity [57,58]. Ang2 was initially identified as a competitive antagonist of the Tie2 receptor, with an affinity comparable to Ang1, effectively blocking Ang1-induced Tie2 phosphorylation [19,59]. However, later studies have shown that Ang2 can induce Tie2 activation in various contexts, including stressed endothelial cells, high Ang2 concentrations, different matrix models, and in the absence of Ang1 [20,21,22,60,61]. One study suggested that Ang2 can act as an autocrine protective factor for stressed endothelial cells [20]. Reduced Akt activity in impaired endothelial cells can be restored through Ang2 induction, which acts as an adaptive mechanism to enhance endothelial cell survival via Tie2-Akt signaling [20].

In a complementary study, another group demonstrated that the exogenous delivery of Ang2 to endothelial cells derived from progeria patients rescues EC dysfunction [17]. That study showed that Ang2 treatment enhanced vasculogenesis by normalizing EC gene expression and migration and also restored nitric oxide bioavailability through eNOS activation [17].

The protective role of Ang2 in that context was also dependent on the Tie2/Akt signaling pathways (Figure 2). Conversely, blocking Ang2 decreases tumor xenograft growth by promoting apoptosis and reducing endothelial cell proliferation [62]. Additionally, Ang2 inhibition using Nesvacumab in control endothelial cells effectively suppressed angiogenesis through Tie2/Akt pathway deactivation. High concentrations of Ang2 (800 ng/mL) act as a pro-survival factor that protects ECs from apoptosis during serum deprivation [21].

Recombinant Ang2 injection into the anterior chamber of rat pups five days after birth stimulated the proliferation of ECs and significantly increased the number of mitotic cells in blood vessels [63]. Another significant response to Ang2 injection was EC migration and sprouting from blood vessels [63]. In a three-dimensional fibrin matrix angiogenesis model, another study found that Ang2, similar to Ang1, can induce an extensive capillary-like tube formation [22]. This effect was accompanied by increased autophosphorylation of the Tie2 receptor in ECs [22].

Since endothelial cells produce Ang2 but not Ang1, it is reasonable to speculate that endogenous Ang2 plays an important role in maintaining Tie2 receptor activity, as well as phosphatidylinositol 3-kinase (PI3K) and Akt signaling [63]. This, in turn, promotes EC survival, migration, and tube formation [63]. All these studies support a model in which Ang2 is important for maintaining proper EC function.

In the context of HGPS, Ang2 may help restore EC function in cases of EC dysfunction and oxidative stress by activating the Tie2 receptor. Another mechanism contributing to vascular abnormality in HGPS is EC senescence and apoptosis. Since Ang2 treatment activates the Tie2/Akt signaling pathway, it may help reduce EC apoptosis and promote survival. By reducing inflammation and improving the mechanosensitivity of HGPS EC affected by progerin, Ang2 treatment may enhance EC’s ability to respond to shear stress, thereby improving overall vascular health. More importantly, due to the crucial role of NO in maintaining vascular function, the rescue effect of Ang2 on NO bioavailability may alleviate some symptoms of EC in HGPS patients. All these improvements in EC may reduce the risk of cardiovascular complications, such as atherosclerosis, making Ang2 treatment a potential therapeutic option for managing atherosclerosis in these patients.

### 2.3. The Possible Role in Valvogenesis and Cardiac Development

In an observational study, 27 patients with classic HGPS mutations were evaluated, and left ventricle (LV) diastolic dysfunction was identified as the most prevalent abnormality [64]. In addition, LV hypertrophy, LV systolic dysfunction, and valve disease tend to become more pronounced with age in these patients [64]. Studies have shown that Ang2 plays a role in embryonic cardiac development and function. Ang2 knockdown (Ang2-KD) mice develop aortic valve stenosis (AVS) at a late embryonic stage [65]. They exhibit a premature thickening of their aortic valve (AoV) leaflet due to an imbalance between cell proliferation, senescence, and apoptosis during AoV remodeling [65]. The observed embryonic stage AVS leads to cardiac dysfunction in adult Ang2-KD mice, characterized by a defect in the left ventricle (LV) [66]. Increased chronic stress associated with NOX4 upregulation in Ang2-KD adult mice causes cellular damage, which further leads to LV systolic dysfunction. This study suggests an important role for Ang2 in maintaining cardiac redox homeostasis [66]. Administration of Ang2-expressing resident cardiac cells in rats after myocardial infarction, on the other hand, improves LV function and increases neovascularization, indicating a potential therapeutic avenue for cardiac repair [67].

Heart development was one of the top 20 functions enriched with *Ang2*-co-expressed genes in functional enrichment analysis of OMICs-based studies [68]. In cardiac allografts, both AAV-Ang1 and AAV-Ang2 reduced inflammation and increased antiapoptotic Bcl-2 mRNA levels and the Bcl-2/Bax ratio after 8 weeks [43]. However, prolonged AAV-mediated Ang1 expression led to smooth muscle cell (SMC) activation, whereas AAV-mediated Ang2 expression did not induce SMC activation and was associated with reduced coronary artery vasculopathy (CAV) [43].

By promoting healthy cardiac remodeling, Ang2 treatment may help rescue some of the cardiac abnormalities seen in HGPS patients, such as LV hypertrophy and systolic dysfunction. In addition, by maintaining the balance between cell proliferation, senescence, and apoptosis, Ang2 may improve HGPS patients’ valve structure and function, which further addresses valve disease in these patients. The role of Ang2 in promoting the formation of new blood vessels, which improves blood supply to damaged tissue, makes it a strong candidate for maintaining cardiac health in HGPS patients after events like myocardial infarction.

### 2.4. The Possible Role in Maintaining Lymphatic Vascular Integrity

In addition to cardiovascular manifestations, HGPS patients’ non-cardiac vessels including lymph nodes, and lymphatic vessels show perivascular adventitial fibrosis. The primary lymphatic organ, the thymus, exhibits reduced size in Lmna^G609G/G609G^ mice [69]. This shrinking is associated with reduced cellularity and increased fibrosis in the thymic vasculature, suggesting a potential reduction in vascularization in this lymphatic organ [69]. Furthermore, the central veins of HGPS patients’ hilar lymph nodes also exhibit extensive perivascular tissue fibrosis [4].

Ang2 plays an important role in lymphatic development in vivo and appears to act as an activating agonist in this situation. Engineered mice lacking Ang2 did not survive the past two weeks of life and were severely compromised by generalized lymphatic dysfunction [70]. Their newborns developed chylous ascites, with the peritoneal cavity filled with a milky fluid [70]. Both large and smaller lymphatic vessels showed abnormal patterning in Ang2-deficient pups [70]. The lymphatic defects were rescued by Ang1 expression; however, vascular remodeling defects were not. Based on these observations, it seems Ang2 plays a key role in the remodeling and maturation of the lymphatics [70].

Complementing this study, another in vivo study indicated Ang2 as a crucial regulator of lymphatic remodeling [71]. Using Ang2-deficient mice, the study showed that these mice have defects in postnatal remodeling and maturation of the lymphatic vasculature [71]. These lymphatic remodeling defects resulted in a profound deficiency of mature collecting lymphatic vessels [71]. They also showed that lack of *Ang2*, caused the premature recruitment of smooth muscle cells to lymphatic vessels, leading to premature stabilization of the lymphatic vasculature and resulting in lymphatic hypoplasia [71].

Lymphatic endothelial cells (LECs) treated with Ang2 showed improved proliferation and survival [72].

These protective effects of Ang2 were mediated through Tie2 binding and activation of the AKT signaling pathway [72]. This study indicated that Ang2 was a more potent anti-apoptotic and proliferative signal than Ang1 in LECs [72]. Ang2 treatment may recover lymphatic vessel defects in progeria, and further help restore the size and functionality of lymphoid organs. Since Ang2 treatments enhanced the survival and proliferation of LECs, it is reasonable to speculate that Ang2 can reduce lymphatic atrophy by increasing lymphoid tissue cellularity. By improving thymic vascularization, Ang2 may support thymocyte development and ultimately create a supportive environment for immune cell development in HGPS. Aside from HGPS, Ang2 may be used for therapeutic purposes in the case of lymphedema or treatment of other clinical disorders of the lymphatic system.

### 2.5. The Possible Role in Promoting White Adipose Tissue Hemostasis

Loss of subcutaneous fat in HGPS patients is one of the major phenotypes that can be detected within the first year of life [30,73]. Lipodystrophy is a disorder that is characterized by partial or total loss of the adipose tissue and has been associated with mutations in the lamin A gene [74,75]. In healthy individuals, fat is stored as a form of triglycerides in white adipose tissue (WAT) [76]. However, in patients with lipodystrophy, the capacity of WAT to store fat is impaired, which results in the accumulation of fat in visceral deposits [77]. This abnormal accumulation and storage result in lipotoxicity, which further increases the risk of cardiovascular disease, hepatic steatosis, and insulin resistance [78].

HGPS patients fail to gain proper weight and lose their body fat despite adequate caloric intake. Studies have shown that mice expressing progerin exhibit adipose tissue depletion, along with increased senescence and inflammation. In addition, HGPS-differentiated adipocytes showed impaired lipid storage capacity [79,80]. Defective AT angiogenesis and disproportional adipocyte hypertrophy reduce AT vasculature, which further leads to fat dysfunction. Using endothelial cell-specific progeroid mice as a model, one study identifies endothelial cell senescence as the cause of impaired adipocyte function and systemic metabolic health observed in HGP [81]. The researcher further concluded that endothelial cells play an important role in maintaining AT health and function through angiocrine factors [81].

Constant vessel remodeling, growth, and regression are required for adult adipose tissue plasticity [82]. Ang2 has been shown to play a protective role in adipose tissue insulin resistance, associated with *obesity*. White adipose tissue (WAT) specific Ang2-expressed mice show improved WAT vascularization and resistance to high-fat diet-induced obesity. This enhancement in metabolic function was associated with a boost in glucose tolerance, insulin sensitivity, and disposal. Mice overexpressing Ang2 showed significant down-regulation of fibrotic gene expression and a decrease in collagen accumulation. These data suggest that in addition to its anti-inflammatory role, Ang2 can also play an anti-fibrotic role [83]. Previous studies have shown that overexpression of vascular endothelial cell growth factor A (VEGF-A) also increases WAT vascularization. However, this improvement was accompanied by a significant increase in vascular permeability [84]. In contrast to the effects of VEGF-A, the induction of Ang2 did not affect vascular permeability [84].

Blocking the Ang2 action in WAT using an anti-Ang2 antibody, on the other hand, increased the level of inflammation and led to fibrosis. This observation suggests the importance of Ang2 in healthy adipose tissue expansion and how defective Ang2 accelerates adipose tissue dysfunction [83]. Previous studies have shown that human FOXC2 mutations are associated with obesity, type 2 diabetes, and defects in lymphangiogenesis [85,86]. Specific over-expression of FOXC2 in adipose tissue converts WAT to brown adipose tissue (BAT) with a four-fold increase in oxygen consumption in transgenic mice (FOXC2-TM) [86]. Using the same transgenic mice model (FOXC2-TM), another group demonstrated that FoxC2 directly controls the promoter activity of Ang2. They showed that the FoxC2-Ang2 signaling is crucial for proper remodeling, patterning, and function and maturation of the adipose tissue. Ang2 expression and regulation are important in adipose tissue vascularization to meet and adapt to certain metabolic demands [87].

Given the beneficial effects of Ang2 in promoting WAT vascularization, it is reasonable to speculate that Ang2 may support adipocyte health and function in HGPS. In addition, Ang2 treatment may improve the impaired AT storage capacity in HGPS patients, consequently improving some of the metabolic complications associated with fat loss seen in these patients. Ang2’s involvement in AT remodeling and maturation also increases the possibility that Ang2 treatment may reverse the detrimental effects of progerin-induced senescence through proper AT development. Beyond HGPS, given the beneficial effects of Ang2 in protecting against insulin resistance and diet-induced weight gain, Ang2 treatment may offer new therapeutic options for the treatment of obesity and metabolic disorders.

### 2.6. The Possible Role in Facilitating Bone Wound Healing

A major medical challenge in HGPS patients is osteoporosis and the high risk of fracture. This condition is characterized by reduced bone mass and density (osteopenia) due to microarchitectural deterioration, resulting in osteoblast dysfunction and increased risk of fractures. In addition to osteoporosis, these patients also exhibit delayed bone healing after fractures, progressive osteolysis, and skeletal dysplasia [88,89]. Generated HGPS mouse models reflect the bone abnormalities observed in the patients, such as growth retardation, joint immobility, skeletal deformation, changes in bone mineral density, and abnormal gait [90,91].

The fractures in HGPS patients are mainly due to an inherent abnormality of the collagen structure or active resorptive process [89]. A sufficient blood supply to the fracture is essential for the proper reconstitution of bone tissue [92]. In an in vivo study, rabbits with radius bone defects, treated with the implantation of a hydroxyapatite/collagen scaffold followed by injections of increasing concentrations of Ang2, showed accelerated repair of the bone defects [93]. Increased expression of the autophagic modulators, such as microtubule-associated protein one light chain 3 (LC3), Beclin-1, and SQSTM1/P62, was detected in the treatment groups. These data suggest that high concentrations of Ang2 improve bone defect repair by activating the autophagy pathway and promoting vascularization [93].

Several studies have identified that improper vascularization or a reduction in vascularization of the regenerative tissue causes impaired bone healing [94,95]. Using a standard mechanically induced delayed healing model in sheep osteotomy, another group demonstrated that Ang2 expression correlates with the rate of bone healing [96]. The extremely low mRNA levels of Ang2 in the delayed healing group compared to the standard group at several time points provide evidence of the importance of Ang2 in the rate of bone healing [96]. The plasma concentrations of Ang2 are correlated with the bone marrow concentrations of Ang2 in leukemia patients, and patients with higher Ang2 levels had longer event-free survival rates [97].

Pulsed electromagnetic field (PEMF) therapy has been widely used to accelerate bone fracture healing [97]. Significant up-regulation of Ang2 induced by PEMF therapy suggests the involvement of this molecule in the bone fracture repair process [97]. This improvement does not have invasive effects, such as inducing hypoxic conditions and inflammation in the bone marrow [97].

The characteristic of Ang2 in enhancing vascularization is crucial for delivering nutrients and oxygen to injured bone tissue, which may help accelerate the bone-healing process in HGPS patients. Additionally, the increased expression of autophagy markers in response to Ang2 treatment suggests that Ang2 may facilitate the clearance of damaged cells through activation of the autophagy pathway, which further promotes tissue regeneration. Ang2 treatment may also provide a more favorable environment for the bone-healing process in HGPS by reducing inflammation. Lastly, by maintaining the balance between bone formation and resorption, Ang2 may help address the osteoporosis pathology seen in these patients.

### 2.7. The Possible Role in Enhancing Blood Flow Recovery in Ischemic Tissue

A histopathology study of two patients with 1824 (C>T) classical HGPS mutation, who died at ages 9.9 and 14.0 of myocardial infarction, revealed advanced atherosclerotic lesions [4]. No sign of thrombus or acute plaque rupture was observed in these regions. However, the presence of healed plaque ruptures increases the likelihood that the atherosclerosis-associated clinical complications seen in HGP resulted from flow-limiting stenosis [4]. Additionally, LmnaG609G knock-in (HGPS) mice exposed to photochemically induced carotid artery endothelial injury exhibited accelerated thrombus formation compared to the WT animals. This resulted in a shorter time to occlusion, which can reduce or even stop the blood flow to downstream tissues, causing ischemic tissue damage. Hind-limb mice receiving HGPS-iPSC-derived mesenchymal stem cells (HGPS-MSCs) immediately after ligation displayed severe limb loss compared to those receiving control mesenchymal stem cells (Con-MSCs) [98]. Out of 15 mice, only 1 mouse showed rescue of the ischemic limb, with significant interstitial fibrosis and extensive muscle degeneration [98]. Using laser Doppler imaging, they monitored blood flow recovery after MSC transplantation and found that the HGPS-MSC group exhibited significantly lower blood flow recovery compared to the Con-MSC groups [98].

Previous studies have shown that hypoxia in endothelial cells, both in vitro and in vivo, induces Ang2 expression [99]. This raises the question of whether Ang2 plays an important role in hypoxia/ischemia-induced vasculogenesis and remodeling. Upregulation of Ang2 in the mouse hindlimb after femoral artery ligation suggests a role of Ang2 in hindlimb ischemia [100]. By contrast, Ang1 mRNA levels were not significantly different between ischemic and non-ischemic conditions [100]. Mice with hindlimb ischemia that were injected with L1-10, an Ang2-specific inhibitor, showed significant impairment in blood flow recovery after femoral artery ligation. Inhibition of Ang2 reduced the coverage of capillary ECs with vascular smooth muscle cells within the ischemic tissue. These data suggest an important role for Ang2 in neovascularization after ischemia.

Previous studies reported that the recovery of ischemic tissue after hindlimb ischemia is achieved through arteriogenesis rather than angiogenesis [101]. This suggests that Ang2 mediates blood flow recovery primarily through arteriogenesis without significantly affecting angiogenesis. Based on these studies, Ang2 treatment may improve the recovery of the ischemic area in HGPS by enhancing atherogenesis, which further increases blood supply. By promoting the interaction between endothelial cells and vascular smooth muscle cells, Ang2 may enhance capillary coverage in HGPS, leading to better structural support for new vessel formation.

### 2.8. The Possible Neuroprotective Effects of Ang2

Even though brain development in HGPS patients is protected from the effects of progerin, they can still experience strokes. In fact, early and clinically silent strokes are one of the major characteristics of the HGPS [102]. Stroke in these patients is revealed as ischemic cerebral infarction due to insufficient cerebral blood supply [103]. The cerebrovascular event can occur in HGPS children at any age, as early as four years old [103]. White matter lesions in the cerebral hemisphere as the result of chronic brain hypoperfusion are also common in these patients [104].

Vascular dysfunction in the brain can disrupt the blood–brain barrier (BBB) and, in turn, disturb cerebral homeostasis [105]. BBB breakdown plays a critical role in the development of neurological disorders, such as neurodegenerative conditions, and, therefore, maintaining its integrity and health is essential [106,107]. High BBB leakage is associated with reduced Ang2 expression [108]. Pericyte-deficient mice with EC-specific *Ang2* knockout show higher levels of BBB leakage compared to the control group. *Ang2* knockout mice also exhibit abnormal vascular morphology accompanied by increased tracer leakage and weak, irregular junctional staining for CLDN5. These observations suggest that Ang2 employs a protective role in maintaining BBB integrity by regulating junctional protein alignment [108].

Studies have shown that pro-angiogenic factors play an essential role in vessel protection, repair, and reconstruction after ischemic stroke. Substantial upregulation of Ang2 mRNA was observed in subventricular zone (SVZ) neural progenitor cells after stroke, suggesting that endogenous Ang2 might be involved in post-stroke neurogenesis. In agreement with this statement, Liu et al. showed that recombinant Ang2 treatments promote the differentiation of neural progenitor cells into the neuronal lineage [109]. This protective effect was Tie2-dependent and associated with up-regulation of transcriptional factor C/EBPβ.

One of the major events that occur after the stroke is the migration of neuroblasts from SVZ toward the ischemic region of the striatum. They also showed that Ang2 treatment mediates the migration of neural progenitor cells, independent of the Tie2 receptor [109]. An in vivo study suggested that exogenous Ang2 plays a protective role during the acute phase of cerebral ischemia. Brain-damaged mice treated with different doses of Ang2 showed a significant dose-dependent decrease in lesion volume compared to the vehicle-treated group [110]. By contrast, timely administration of VEGF increased ischemia-induced vascular permeability and intensified ischemic damage. However, Ang2 was able to reverse the detrimental effects of VEGF on lesion size and vascular permeability [110]. In agreement with this study, Lv et al. demonstrated that Ang2 administration decreases infarction size and reduces neuronal loss after brain injury by promoting angiogenesis [111]. Exogenous Ang2 induces angiogenesis by upregulating CD34 expression in cultured ECs in vitro and extending the CD34-positive vascular area and length during the acute phase of ischemia. Injection of Ang2 into the striatum of a cerebral ischemic injury mouse model reduced the volume of cerebral infarction and increased the length of CD34-positive blood vessels per unit area [111].

All these observations confirm the promotional effects of Ang2 on angiogenesis after ischemic stroke. Given its vaso-protective, as well as neuroprotective, effects during cerebral ischemia, Ang2 may represent a potential target for treatment. Since there are no reports of revascularization or thrombolysis in HGPS patients who have experienced cerebral infarction, Ang2 treatment may be considered a potential preventive approach. Given the involvement of Ang2 in post-stroke neurogenesis, this treatment may enhance and facilitate brain tissue recovery and regeneration after ischemic events in these patients. Furthermore, by inducing the migration of neuroblasts to the ischemic area, Ang2 may aid in the repair process, which potentially leads to better outcomes. The pivotal functions and roles of Ang2 in various biological contexts are summarized in Table 1.

## 3. Risks and Challenges of Ang2 Therapy

### 3.1. Context-Specific Challenges

Given the context-dependent activity of ang2, targeting it still carries risks despite its therapeutic potential. One major concern is the role of Ang2 in **vascular instability**, particularly **by increasing EC permeability**. Under certain physiological conditions, Ang2 acts as an antagonist in blood endothelial cells (BECs) and inhibits Ang1-induced Tie2 phosphorylation [19]. The destabilizing effect of Ang2 on ECs was demonstrated in a 3D co-culture model of ECs and VSMCs, where Ang2 treatment disrupted EC integrity [113]. Furthermore, inflammation and angiogenesis are two correlated conditions, and Ang2 has been shown to be involved in inflammatory diseases such as autoimmune diseases, sepsis, and acute lung injury [114]. Serum levels of Ang2 are associated with other inflammatory markers, such as white blood cell count and high C-reactive protein (CRP), suggesting that this molecule may function as a potential cytokine [115].

Therefore, loss of vessel integrity is a significant concern with Ang2 treatment that needs to be carefully considered and evaluated before drawing definitive conclusions about its therapeutic potential. However, the context-dependent activity of Ang2 offers hope for positive outcomes in progeria, although much more investigation is needed.

Uncontrolled angiogenesis is another risk associated with Ang2 treatment, which may even worsen some pathological conditions. Studies on tumor angiogenesis and metastasis have been suggesting an important role of Ang2 in cancer. The Ang2 levels are associated with the progression of small- and non-small-cell lung cancers. Furthermore, EC-specific Ang2 overexpression increases metastasis in lung cancer. This lymphogenic effect is blocked by inhibiting Ang2 function using an anti-Ang2 monoclonal antibody [116]. The Ang2 levels negatively correlate with overall survival in patients with liver cancer [117]. Additionally, blocking Ang2 expression in cervical cancer cells decreases metastasis, which is caused by a reduction in microvessel density [118].

Given the strong evidence that Ang2-mediated signaling increases the risk of tumor progression and metastasis, its therapeutic potential must be carefully considered. Even though improving angiogenesis confers beneficial effects in the case of vascular pathology associated with progeria, uncontrolled angiogenesis also increases the risk of developing tumors and causing cancer. However, it is important to note that, despite all the pathological issues observed in progeria patients, cancer has never been reported in these patients. This could be due to defective angiogenesis, which limits the potential of cancer growth, suggesting the risk of inducing cancer may not be as pronounced in this context. Nevertheless, further and deeper investigation is essential to fully understand the balance between Ang2’s improvement of angiogenesis and uncontrolled angiogenesis.

### 3.2. Clinical Challenges

Even though Ang2 therapies seem promising in in vivo models, several clinical challenges need to be addressed before these therapies can be translated into humans. One of the major challenges is determining the optimal dosage and the effective delivery methods for such therapies. Conducting dose–response studies in a progeria mouse model will help determine the most effective and safe concentration of Ang2 that provides endothelial restoration without causing off-target effects. Testing different delivery methods, such as recombinant Ang2, Ang2 analogs, or Ang2 adenovirus or lentiviral systems, in a progeria mouse model would be beneficial for identifying the most efficient and targeted approach to maximize its therapeutic efficacy. Each therapeutic option has its own challenges. For example, while recombinant Ang2 appears to be more effective due to its ability to oligomerize quickly and activate Tie2 at lower doses, its short half-life (approximately 18 h) requires daily injections, making this option more labor-intensive. By contrast, a single systemic administration of Ang2 adenovirus, delivered directly into the tail vein of 8-week-old apoE−/− mice (atherosclerosis-prone), seems to reduce atherosclerotic lesions [36]. However, the long-term effects of this treatment still need to be evaluated. Conducting a longitudinal study on progeria mice receiving intravenous administration of Ang2 adenovirus would help address potential long-term safety concerns. Another important consideration for viral delivery is the tropism of viral transduction, which helps determine the most suitable viral vector to efficiently deliver Ang2 to endothelial cells without infecting non-target cells or triggering unwanted immune responses. The potential tumorigenic effects of Ang2 due to abnormal angiogenesis effects also need to be monitored.

Choosing both young and old progeria mice would help determine the impact of timing on therapeutic potential, as well as identify how factors such as age and disease stage influence treatment outcomes. It is also important to note that while Ang2 seemed to reverse the toxic effects of progerin in endothelial cells, it does not affect progerin expression itself. Therefore, exploring combination therapies would be a promising approach, including testing whether Ang2 treatment synergizes with other potential therapies for HGPS, such as farnesyl transferase inhibitors (FTIs), senolytics, gene therapy, rapamycin, and telomere therapy.

## 4. Summary

In this review, we discussed the potential, multi-layered protective effects of Ang2 across various physiological and pathological contexts and demonstrated its possible benefits in treating HGPS. We emphasized the role of this molecule as more than just a Tie2 antagonist and showed that it also exhibits context-dependent agonistic properties that contribute to significant protective effects, including enhancement of vascular integrity, promotion of tissue repair, and modulation of inflammatory responses. These properties are particularly relevant in the context of HGPS, where progressive atherosclerosis, cardiovascular complications, and stroke present significant challenges in these patients. The beneficial effects of Ang2 arise from its ability to activate the Tie2 receptor, which in turn activates PI3K/AKT pro-survival pathways. The studies presented in this paper indicate that the protective effects of Ang2 are not limited to vascular health alone (Figure 3). In the context of atherosclerosis and cardiovascular disease, Ang2 treatment seems to significantly reduce lesion size and inhibit LDL oxidation through eNOS activation. This NO-dependent mechanism highlights the important role of Ang2 in attenuating oxidative stress and inflammation, which are responsible for the formation and progression of atherosclerotic plaque. All this evidence supports the idea that Ang2 may play a multi-layered role in protecting the vasculature of HGPS patients against accelerated atherosclerosis. The therapeutic effect of Ang2 in restoring EC function and promoting vascular health further validates its atheroprotective role in this context. Additionally, its protective role in cardiac function, tissue repair, and inhibition of inflammation and apoptosis makes Ang2 a strong candidate for supporting cardiac health in HGPS patients (Figure 3). In the lymphatic system, Ang2 maintains lymphatic integrity through vessel maturation and remodeling, reducing fibrosis and promoting cellular proliferation. These improvements may help reverse the lymphoid atrophy observed in these patients (Figure 3).

Additionally, the ability of Ang2 to promote WAT vascularization highlights its potential as a therapeutic agent for improving lipodystrophy in HGPS patients. Ang2’s protective effect in tissue repair and bone wound healing may help address the osteoporosis observed in these patients. Furthermore, Ang2 can enhance brain vascular integrity and promote neurogenesis associated with ischemic injury, which emphasizes its neuroprotective role. Based on this, Ang2 treatment may also enhance cerebrovascular health in these patients by supporting stroke prevention or recovery after ischemia (Figure 3).

Translating these findings into clinical practice, however, requires further investigation. The context-dependent impacts of Ang2 must be carefully considered to avoid potential adverse effects such as the risk of unwanted vessel growth or instability. Future studies should focus on defining the optimal conditions for Ang2-based therapies and exploring their long-term safety and efficacy.

## Figures and Tables

**Figure 1 ijms-25-13537-f001:**
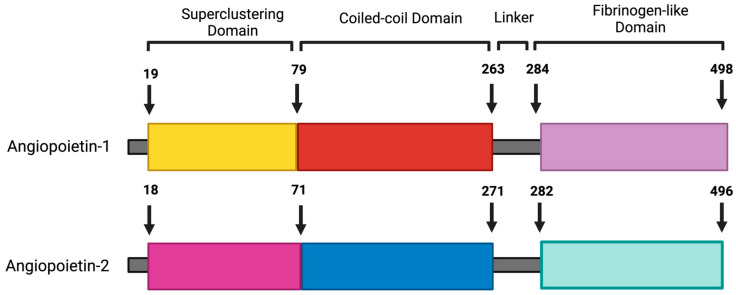
Schematic representation of the protein structures of native Ang1 and Ang2. This diagram illustrates the structural components of both Ang1 and Ang2, highlighting key domains and motifs. Each protein is characterized by an N-terminal signal peptide that directs the protein to its secretion pathway. Following the signal peptide, both Ang1 and Ang2 contain a superclustering domain, which facilitates the dimerization of these proteins. This is followed by a coiled-coil domain, important for protein–protein interactions, and short linkers that connect the various functional domains. The C-terminal fibrinogen-like receptor-binding domain is responsible for binding to Tie2 receptors on endothelial cells. This figure was created using BioRender.

**Figure 2 ijms-25-13537-f002:**
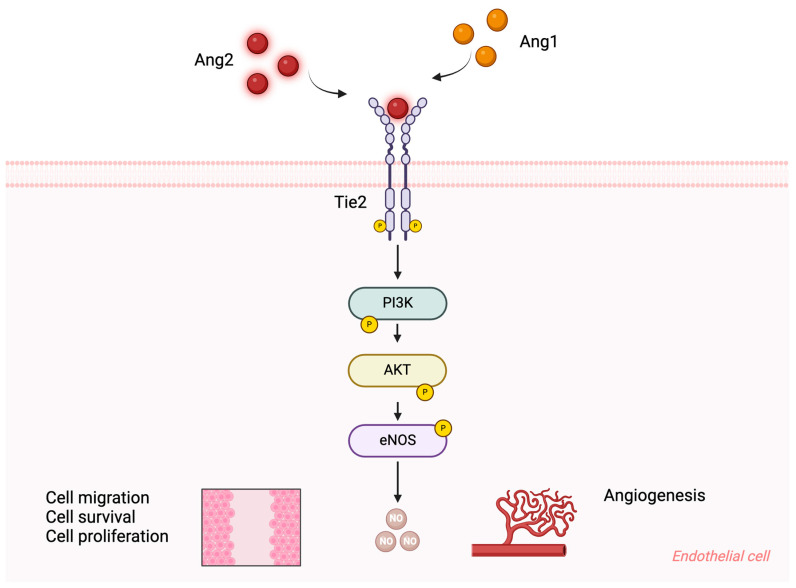
Interaction of angiopoietins with the Tie2 receptor. This schematic illustrates the binding of Angiopoietin-1 (Ang1) and Angiopoietin-2 (Ang2) to the Tie2 receptor located on endothelial cells (ECs). Ang1 and Ang2 act as Tie2 agonists, promoting Tie2 receptor phosphorylation, which activates downstream signaling pathways, including PI3K and Akt. These signaling events play a crucial role in EC processes such as migration, proliferation, and survival, supporting angiogenesis and maintaining vascular integrity. Upon Tie2 phosphorylation, Akt activation also enhances the production of nitric oxide (NO), which contributes to vascular tone regulation and endothelial function, further supporting the homeostasis of the vascular system. This figure was created using BioRender.

**Figure 3 ijms-25-13537-f003:**
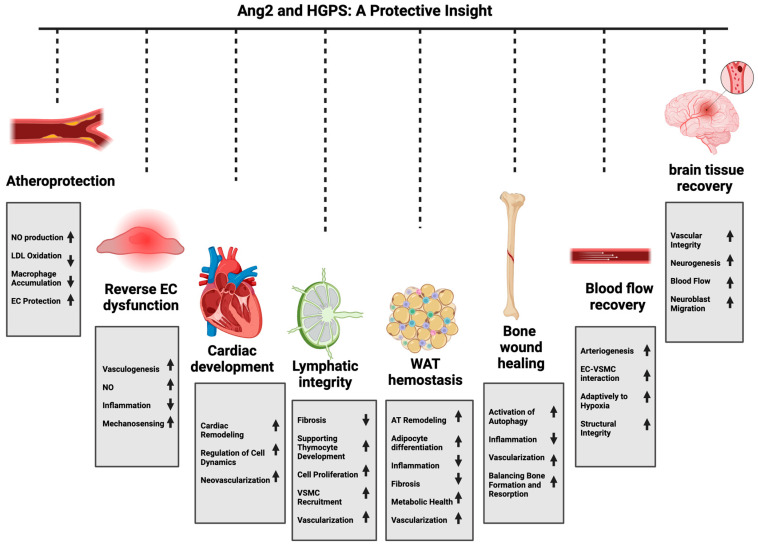
Schematic representation highlighting the potential protective roles of Ang2 in HGPS across various tissues. This figure illustrates the diverse protective effects of Ang2 in multiple tissues affected by HGPS. Ang2 may exert atheroprotective effects by increasing NO production, reducing LDL oxidation, and decreasing macrophage accumulation. In the context of endothelial dysfunction, Ang2 has the potential to reverse dysfunction through increased vasculogenesis, improved NO mechanosensing production, and reduced inflammation. In the cardiovascular system, Ang2 may improve cardiac remodeling and cellular dynamics, promoting neovascularization and better regulation of heart development. Furthermore, Ang2 could enhance lymphatic integrity by reducing fibrosis, supporting thymocyte development, and improving cell proliferation, vascular smooth muscle cell (VSMC) recruitment, and vascularization. In white adipose tissue (WAT), Ang2 may improve homeostasis through adipose tissue remodeling, adipocyte differentiation, and enhancing metabolic health, while reducing fibrosis and inflammation. For bone tissue, Ang2 has the potential to promote wound healing by activating autophagy, reducing inflammation, and improving vascularization, which supports the balance between bone formation and resorption. Additionally, Ang2 can enhance blood flow recovery by promoting arteriogenesis and improving the interaction between ECs and VSMCs, strengthening structural integrity and adaptability to hypoxic conditions. In the brain, Ang2 has been shown to improve vascular integrity, increase neurogenesis, promote neuroblast migration, and enhance blood flow, supporting tissue recovery in neurovascular pathology. This figure was created using BioRender.

**Table 1 ijms-25-13537-t001:** This table provides a detailed summary of the pivotal functions and roles of Ang2 in various biological contexts. It outlines the evidence supporting each function, the underlying mechanisms involved, and the associated outcomes. Each entry is supported by relevant references.

Function/Role	Evidence	Mechanism	Outcome	References
Atheroprotection	Reduction in atherosclerotic lesion size, macrophage accumulation, and LDL oxidation	eNOS activation	Protects against atherosclerosis by modulating LDL oxidation and inflammation	[17,36,44]
Alleviating Endothelial Dysfunction	Enhancement of EC survival, migration, and tube formation	Tie2-Akt signaling pathway	Promotes endothelial cell function and angiogenesis	[17,22,58,63]
Cardiac Development and Function	Aortic valve stenosis and cardiac dysfunction in Ang2 knockdown mice. Improvement of heart function upon Ang2 administration	Maintains cardiac redox balance and promotes neovascularization	Ang2 is crucial for cardiac development and function	[65,67,68,112]
Lymphatic Vascular Integrity	Severe lymphatic dysfunction, abnormal vessel patterning, and chylous ascites in Ang2 deficient mice	Tie2 activation	Ang2 is essential for lymphatic vessel development and function	[70,71,72]
White Adipose Tissue Homeostasis	Improvement in WAT vascularization, glucose tolerance, and insulin sensitivity in Ang2 overexpressed mice	Enhances WAT vascularization and exerts anti-fibrotic effects	Protects against obesity-related inflammation and fibrosis	[83,84,86]
Bone Wound Healing	Accelerated bone defect repair in rabbits upon Ang2 treatment	Autophagy activation	Enhances bone healing and vascularization in bone defect	[93,96]
Blood Flow Recovery in Ischemia	Upregulation of Ang2 in hindlimb ischemia models aids in blood flow recovery	Promotes arteriogenesis and capillary coverage	Improves blood flow recovery and neovascularization post-ischemia	[100]
Neuroprotection	Ang2 improves BBB integrity and promotes neurogenesis post-stroke	Promotes vascular integrity, supports neurogenesis, and enhances angiogenesis	Protects BBB integrity and promotes recovery after ischemic stroke	[108,109,110,111]

## Data Availability

No new data were created or analyzed in this study. Data sharing does not apply to this article.

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
