# Peer review of "Angiopoietin-2: A Therapeutic Target for Vascular Protection in Hutchinson–Gilford Progeria Syndrome"

_ijms, 2024, doi:10.3390/ijms252413537_

Round 1
Reviewer 1 Report
Comments and Suggestions for Authors
This manuscript effectively highlights Ang2’s therapeutic promise for HGPS-associated vascular dysfunction, offering valuable insights into its mechanistic roles. However, addressing the outlined weaknesses will enhance its clarity, applicability, and scientific rigor, strengthening its impact on the field.
Recommendations
Contextual Limitations:
The manuscript focuses on Ang2’s benefits but overlooks its potential risks and context-specific challenges.
Suggestion: Discuss Ang2’s dual roles as both agonist and antagonist, addressing risks like vessel instability and unwanted angiogenesis.
Mechanistic Depth:
Pathways such as Tie2/AKT are briefly mentioned but lack detailed exploration across tissues.
Suggestion: Expand on downstream signaling pathways and their specific roles in HGPS and normal physiology.
Narrow Focus on Ang2:
Other angiogenic factors, like VEGF, are underexplored, limiting a broader therapeutic perspective.
Suggestion: Include a comparative analysis of Ang2 and alternative targets, emphasizing their strengths and weaknesses.
Translational Gaps:
Practical aspects of implementing Ang2 therapy, such as dosage and delivery, are missing.
Suggestion: Address clinical challenges, including long-term safety and efficacy considerations.
Visual Improvements:
Figures and tables are informative but could be clearer.
Suggestion: Add detailed legends and annotations to enhance understanding.
iThenticate report shows 24% match. Please improve it.
Reviewer 2 Report
Comments and Suggestions for Authors
The authors present an in-depth coverage of the clinical manifestation of Hutchinson-Gilford Progeria Syndrome (HGPS) and the therapeutic potential of angiopoietin 2-orientated therapies and treatment strategies in the context of this disease. Briefly, HGPS has been shown to present with vascular complications that sees those presenting with such live into adolescence and not beyond. The vascular stabilising effect of angiopoietin 2 has been earmarked as potentially having an impact in ablating the effects of this disease reducing the risk of mortality and improving the quality of life for those patients receiving such. Throughout the review, the authors highlight the role progerin plays on various physiological systems, while also highlighting the impact of angiopoietin 2 in each respective system, and potential interplay both molecules may have. Overall this was a well-constructed, informative review that delivers detail in a concise and clear form for the most part, while not shying away from the depth this area has.
In reviewing the manuscript I made a few observations. The following should be considered by the authors when preparing a suitable revision.
1. The resolution/formatting of Figure 3 could be improved. At present the text is incredibly difficult to read in the grey boxes, and I would also question whether there is any intention in how the boxes are organised with respect to each other i.e. is there a reason ‘atheroprotection’ is ‘higher’ than ‘lymphatic integrity’?
2. The inclusion of the table is appreciated, and is additive to the text in summarising the key impacts of Ang2 in each biological axis. I would recommend however the formatting of this table be reviewed as the formatting in terms of paragraphs breaks etc. makes the layout difficult to read.
3. Overall the quality of the writing is very good, and the authors have invested a clear effort in trying to capture the broad scope of Ang2 treatment across the different physiological systems. In reviewing each section though I would recommend that paragraph breaks be inserted in order to ‘break’ the text up somewhat. Some sections are quite dense in terms of detail, and I feel strategic breaks in the text will improve the flow of the piece overall.
